# Mixed cytomegalovirus genotypes in HIV-positive mothers show compartmentalization and distinct patterns of transmission to infants

Juanita Pang[1†], Jennifer A Slyker[2†], Sunando Roy[1], Josephine Bryant[1], Claire Atkinson[3], Juliana Cudini[1], Carey Farquhar[4], Paul Griffiths[3], James Kiarie[5], Sofia Morfopoulou[1], Alison C Roxby[4], Helena Tutil[1], Rachel Williams[1], Soren Gantt[6], Richard A Goldstein[1‡], Judith Breuer[7‡*]

[1]Division of Infection and Immunity, University College London, Cruciform Building, London, United Kingdom; [2]Departments of Global Health and Epidemiology, University of Washington, Seattle, United States; [3]Institute of Immunology and Transplantation, Division of Infection and Immunity, University College London, London, United Kingdom; [4]Departments of Global Health, Epidemiology, Medicine (Div. Allergy and Infectious Diseases), University of Washington, Seattle, United States; [5]University of Nairobi, Department of Obstetrics and Gynaecology, World Health Organization, Nairobi, Kenya; [6]Research Centre of the Sainte-Justine University Hospital, Department of Microbiology, Infectious Diseases and Immunology, University of Montréal QC, Montréal, Canada; [7]Department of Infection, Immunity and Inflammation, UCL Great Ormond Street Institute of Child Health, University College London, London, United Kingdom

*For correspondence:
j.breuer@ucl.ac.uk

†These authors contributed equally to this work
‡These authors also contributed equally to this work

Competing interests: The authors declare that no competing interests exist.

**Abstract** Cytomegalovirus (CMV) is the commonest cause of congenital infection and particularly so among infants born to HIV-infected women. Studies of congenital CMV infection (cCMVi) pathogenesis are complicated by the presence of multiple infecting maternal CMV strains, especially in HIV-positive women, and the large, recombinant CMV genome. Using newly developed tools to reconstruct CMV haplotypes, we demonstrate anatomic CMV compartmentalization in five HIV-infected mothers and identify the possibility of congenitally transmitted genotypes in three of their infants. A single CMV strain was transmitted in each congenitally infected case, and all were closely related to those that predominate in the cognate maternal cervix. Compared to non-transmitted strains, these congenitally transmitted CMV strains showed statistically significant similarities in 19 genes associated with tissue tropism and immunomodulation. In all infants, incident superinfections with distinct strains from breast milk were captured during follow-up. The results represent potentially important new insights into the virologic determinants of early CMV infection.

## Introduction

Human cytomegalovirus (CMV) is the commonest infectious cause of congenitally acquired disability (*Morton and Nance, 2006*). Between 0.2% and 2% of all live births have congenital CMV infection (cCMVi), and of these, an estimated 15–20% develop permanent sequelae ranging from sensorineural hearing loss to severe neurocognitive impairment (*Boppana et al., 2013*; *Dollard et al., 2007*). Maternal coinfection with HIV, even when mitigated by antiretroviral treatment, is associated with

higher CMV viral loads in plasma, saliva, cervix, and breast milk, and a greater risk of both congenital and postnatal CMV transmission (*Gantt et al., 2016a*; *Gantt et al., 2016b*; *Slyker et al., 2017*; *Richardson et al., 2016*). Numerous studies have highlighted the negative health impacts of CMV on both HIV-infected and HIV-exposed uninfected (HEU) infants and children (*Garcia-Knight et al., 2017*; *Gompels et al., 2012*; *Hsiao et al., 2013*).

Primary maternal CMV infection during pregnancy confers a 30–40% risk of transmission to the foetus (*Kenneson and Cannon, 2007*). Pre-existing maternal CMV immunity appears to reduce the risk of cCMVi, though it is clearly imperfect (*Britt, 2017*). Over two-thirds of infants with cCMVi are born to seropositive women, which constitute 88.4% of women in the Kenyan community from whom these study participants were drawn (*Maingi and Nyamache, 2014*). Moreover, the overall risk of cCMVi is directly proportional to the maternal seroprevalence in a population (*de Vries et al., 2013*). Increasing evidence points to the importance of maternal CMV reinfection with new antigenic strains during pregnancy as a major risk factor for non-primary cCMVi (*Britt, 2017*; *Boppana et al., 1999*). Evidence that household children may be a source of maternal reinfection provides additional support for this hypothesis (*Boucoiran et al., 2018*; *Barbosa et al., 2018*).

The CMV genome is the largest of the human herpesviruses. Regions of extensive sequence variability, together with high levels of recombination between different strains, result in high diversity for a DNA virus (*Lassalle et al., 2016*; *Pokalyuk et al., 2017*; *Sackman et al., 2018*). Individuals are often infected with multiple CMV strains. We have recently demonstrated that separate CMV haplotypes can be resolved from high-throughput sequencing data (*Cudini et al., 2019*). This advance, by enabling tracking of individual genomes within mixed CMV infections, has already revealed the impact of mutation, recombination, and selection in shaping the course of infection (*Cudini et al., 2019*). Here we apply these methods to CMV genomes sequenced from samples from five HIV-infected Kenyan women and their infants that were collected between 1993 and 1998 originally for studies of maternal–infant HIV transmission (*Richardson et al., 2016*). By reconstructing genome-wide haplotypes from these longitudinal samples, we are able to examine the diversity of CMV shed by HIV-infected women and the specific genotypes that are transmitted in congenital and postnatal infections, and to reconstruct the likely chronology with which specific CMV variants were transmitted from mothers to infants.

## Results

### Participant characteristics, sampling, and depth of sequencing

Details of the study cohort, follow-up, sample collection, and HIV and CMV infection status and transmission have been previously described (*Drake et al., 2012*; *Roxby et al., 2014*; *Slyker et al., 2014*). Sufficient residual sample was available from the five families analysed here. To maximize the chance of recovering near full genomes, we selected samples reported in the original publication (*Roxby et al., 2014*) to have $>10^3$ copies/ml, as this is the limit at which we generally can generate whole genomes from blood. Of the five mother–infant pairs analysed, four infants were HEU (infants 22, 123, 41, 14), and one was HIV infected (infant 12).

### CMV viral loads and sequencing

Cervical, breast milk, and infant blood CMV viral loads; mother blood plasma HIV viral loads; and time of sample collection for the five mother–infant pairs studied are shown in *Figure 1*. The percentage of genome coverage and mean read depths are shown in *Table 1*. While breast milk samples had greater than 70% coverage at depths of 10× or more, the cervical and infant samples were generally of lower depth, likely due to degradation of DNA due to the age and handling of the samples; genome coverage and mean de-duplicated read depth were directly related to actual CMV genome copy number present in the input material (*Figure 1—figure supplement 1*). For all subsequent analysis, we removed samples with genome coverage of less than 20%. Fourteen of the remaining 20 cervical and baby samples had genome coverage above 70% and read depths of greater than 10× (*Table 1*).

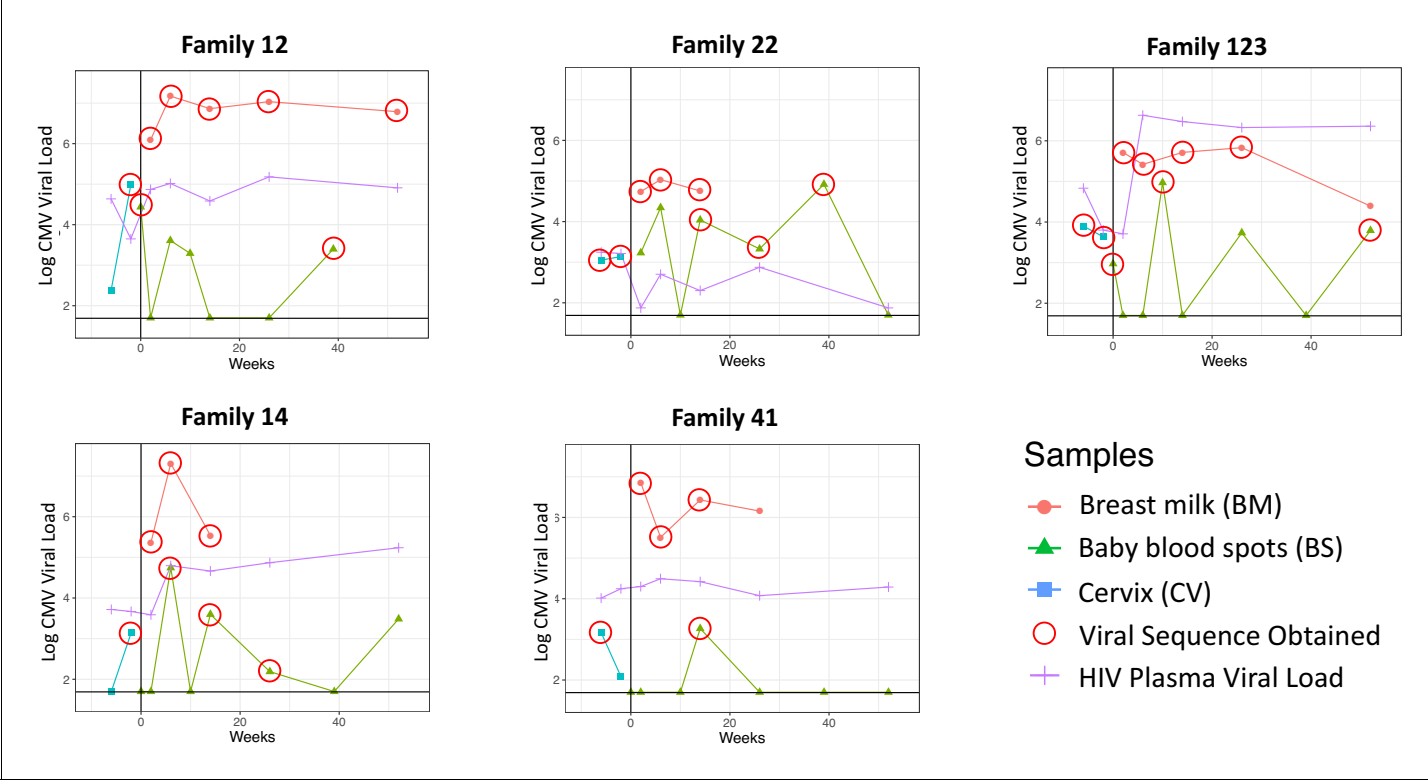

**Figure 1.** Cytomegalovirus (CMV) viral loads of longitudinal samples for each family from breast milk (red), baby blood spots (green), and cervix (blue), and HIV viral loads from mother's blood plasma. Vertical line indicates date of delivery. Horizontal line indicates minimum threshold of detection. Red circles indicate the samples that were submitted for whole-genome sequencing.

The online version of this article includes the following figure supplement(s) for figure 1:

**Figure supplement 1.** Scatter plots showing relationship between input viral load and (A) mean read depth and (B) genome coverage, respectively.

## CMV genome sequence relatedness and diversity

We used multidimensional scaling to cluster CMV genomic sequences by nucleotide similarity (*Figure 2*), as use of phylogenetic trees is problematic due to the high levels of CMV recombination. Sequences from families 12, 14, and 41 all clustered by family. Families 22 and 123 clustered in two distinct spaces, suggesting infection with more than one strain. In all five cases, the first sample from each infant (indicated by an arrow) clustered most closely with that of its mother, indicating the likelihood of recent maternal–infant transmission.

To further investigate the possibility of mixed infections, we calculated the within-sample nucleotide diversity, a metric that we have shown previously can be used as a proxy for the likelihood of mixed strain infections (*Cudini et al., 2019*). It has previously been reported that a nucleotide diversity of 0.005 or above is likely to indicate a mixed infection (*Cudini et al., 2019*). *Figure 2—figure supplement 1* shows that almost all the breast milk samples were highly diverse and therefore likely to contain multiple virus strains, a finding consistent with previous analyses of breast milk from HIV-infected women (*Suárez et al., 2019*). In contrast, the cervical and infant samples, with the exception of one cervical sample from family 12, showed lower diversity. We used subsampling to demonstrate that computed nucleotide diversities are robust down to sequencing depths of >10 (*Figure 2—figure supplement 2*). Low diversity was also observed in cervical and blood spots with higher coverage and read depths (*Table 1*).

## Reconstruction of individual haplotypes reveals CMV compartmentalization

To resolve the individual viral sequences (haplotypes) within each sample, we used our previously described method HaROLD (*Pang et al., 2020a*). *Figure 3* shows that haplotypes for each sample

Table 1. Sequencing characteristics for samples from each family.

% OTR, % of on target read; % Genome, % of genome coverage; % Dup, % of duplicated reads. Samples with genome coverages too low to be included in any analysis are shaded in grey. Cervical or baby samples with good coverage and read depth are highlighted in yellow.

| Sample | % OTR | % Genome | % Dup | Mean depth | Viral load |
|---|---|---|---|---|---|
| **Family 12** | | | | | |
| Breast milk 2W | 26.41 | 99 | 29.49 | 224.45 | 1,235,136.63 |
| Breast milk 6W | 68.99 | 99 | 13.84 | 578.56 | 14,926,741 |
| Breast milk 14W | 76.4 | 99 | 5.02 | 683.04 | 7,309,960 |
| Breast milk 6M | 77.47 | 99 | 8.07 | 730.04 | 10,876,521 |
| Breast milk 12M | 77.81 | 99 | 7.68 | 779.72 | 6,135,712.5 |
| Cervix 38W pregnant | 14.73 | 99 | 47.56 | 325.97 | 95,842 |
| Baby delivery | 1.35 | 76 | 82.27 | 31.86 | 27,393.9395 |
| Baby 6W | 0.02 | 2 | 81.79 | 0.29 | 4067.86694 |
| Baby 10W | 0.1 | 12 | 77.77 | 2.63 | 1959.9679 |
| Baby 9M | 1.1 | 78 | 79.41 | 28.53 | 2501.75195 |
| **Family 14** | | | | | |
| Breast milk 2W | 13.54 | 98 | 65.41 | 101.66 | 232,442.219 |
| Breast milk 6W | 60.32 | 98 | 49.85 | 656.47 | 20,485,190 |
| Breast milk 14W | 11.15 | 97 | 65.77 | 80.09 | 345,851.781 |
| Cervix 38W pregnant | 0.22 | 63 | 56.04 | 4.34 | 1377 |
| Baby 6W | 1.4 | 91 | 69.35 | 21.35 | 55,400.7148 |
| Baby 14W | 3.33 | 96 | 78.59 | 113.92 | 3960.64233 |
| Baby 6M | 0.34 | 66 | 74.11 | 11.42 | 154.414169 |
| Baby 12M | 0.02 | 7 | 75.97 | 0.75 | 3054.47485 |
| **Family 22** | | | | | |
| Breast milk 2W | 6.08 | 96 | 34.22 | 54.34 | 55,000.2891 |
| Breast milk 6W | 43.18 | 98 | 44.57 | 352.49 | 107,861.141 |
| Breast milk 14W | 6.4 | 97 | 44.41 | 38.3 | 56,883.9805 |
| Cervix 34W pregnant | 0.16 | 46 | 54.95 | 2.97 | 1125 |
| Cervix 38W pregnant | 0.16 | 67 | 47.91 | 4.14 | 1377 |
| Baby 2W | 0.01 | 1 | 46.34 | 0.03 | 1703.49292 |
| Baby 6W | 0.08 | 1 | 43.61 | 0.03 | 22,082.6465 |
| Baby 14W | 2.29 | 92 | 79.42 | 46.53 | 10,962.7197 |
| Baby 6M | 0.3 | 33 | 79.36 | 5.98 | 2124.86548 |
| Baby 9M | 0.22 | 25 | 79.33 | 5.01 | 82,937.5 |
| **Family 41** | | | | | |
| Breast milk 2W | 43.33 | 98 | 60.89 | 224.53 | 7,163,743 |
| Breast milk 6W | 37.05 | 98 | 61.89 | 289.61 | 323,325.531 |
| Breast milk 14W | 48.15 | 98 | 68.02 | 438.05 | 2,697,832.75 |
| Cervix 38W pregnant | 0.61 | 91 | 47.53 | 12.6 | 122 |
| Baby 14W | 0.12 | 32 | 74.47 | 4.67 | 1848.62402 |
| **Family 123** | | | | | |
| Breast milk 2W | 16.11 | 98 | 60.11 | 117.25 | 518,071.875 |
| Breast milk 6W | 16.96 | 98 | 64.77 | 107.35 | 262,400.719 |
| Breast milk 14W | 13.95 | 98 | 64.01 | 122.08 | 518,071.875 |
| Breast milk 6M | 15.81 | 98 | 63.07 | 101.92 | 678,250.313 |

*Table 1 continued on next page*

*Table 1 continued*

| Sample | % OTR | % Genome | % Dup | Mean depth | Viral load |
|---|---|---|---|---|---|
| Cervix 34W pregnant | 2.45 | 97 | 49.46 | 41.91 | 7931 |
| Cervix 38W pregnant | 1.36 | 96 | 49.61 | 28.07 | 4326 |
| Baby delivery | 0.21 | 84 | 10.93 | 6.1 | 939.190735 |
| Baby 10W | 2.19 | 91 | 78.64 | 43.96 | 93,297.3047 |
| Baby 6M | 0.13 | 20 | 77.67 | 3.1 | 5428.83545 |
| Baby 12M | 1.36 | 85 | 80.13 | 40.56 | 6205.88281 |

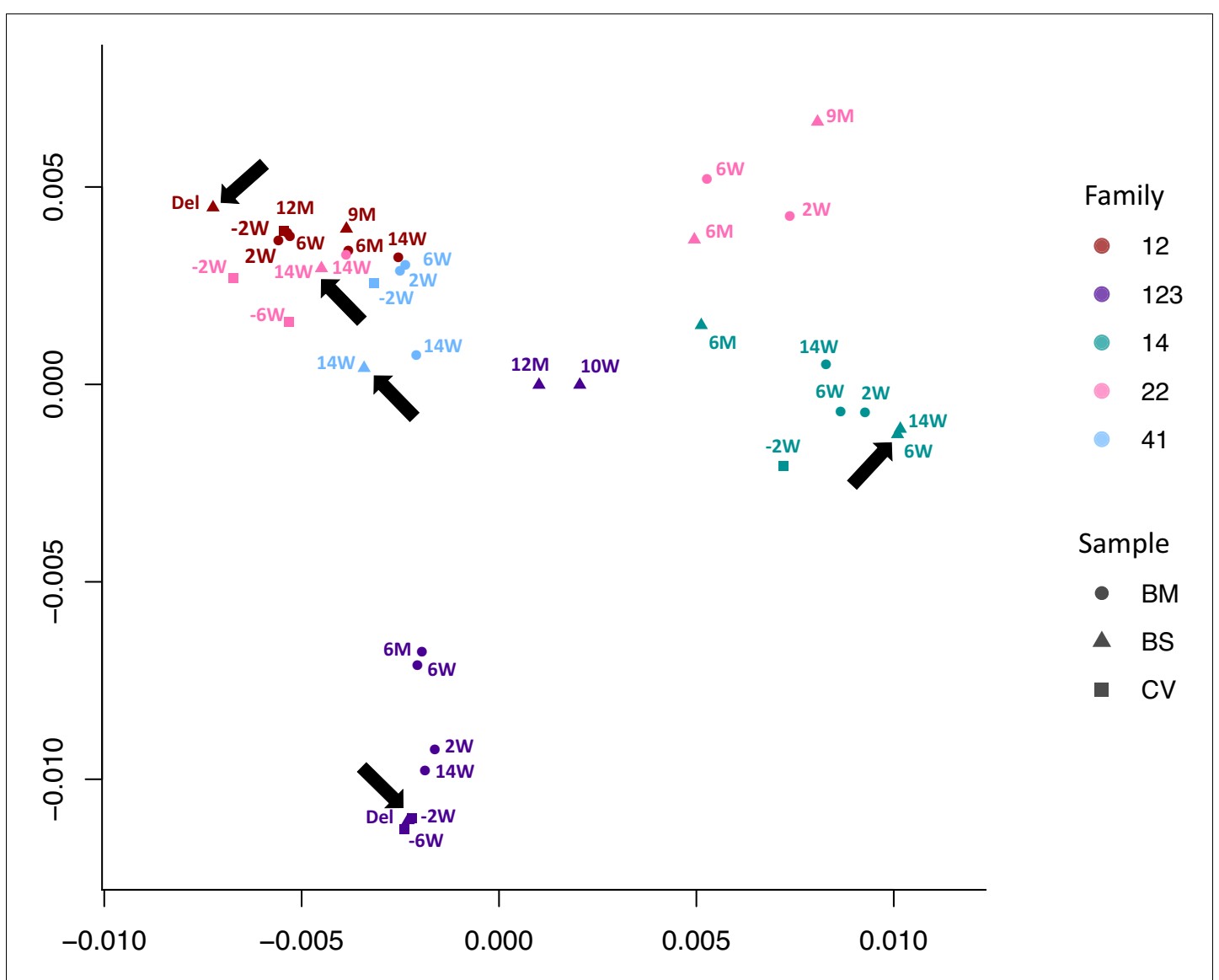

**Figure 2.** Multidimensional scaling showing clustering of consensus genome sequences for each sample by family. Arrows indicate that the first baby blood spot clusters with their own maternal sequences in all cases.

The online version of this article includes the following figure supplement(s) for figure 2:

**Figure supplement 1.** Within sample nucleotide diversity shown by family (colour) and sample type (symbol).

**Figure supplement 2.** Effect of down-sampling on estimated diversity.

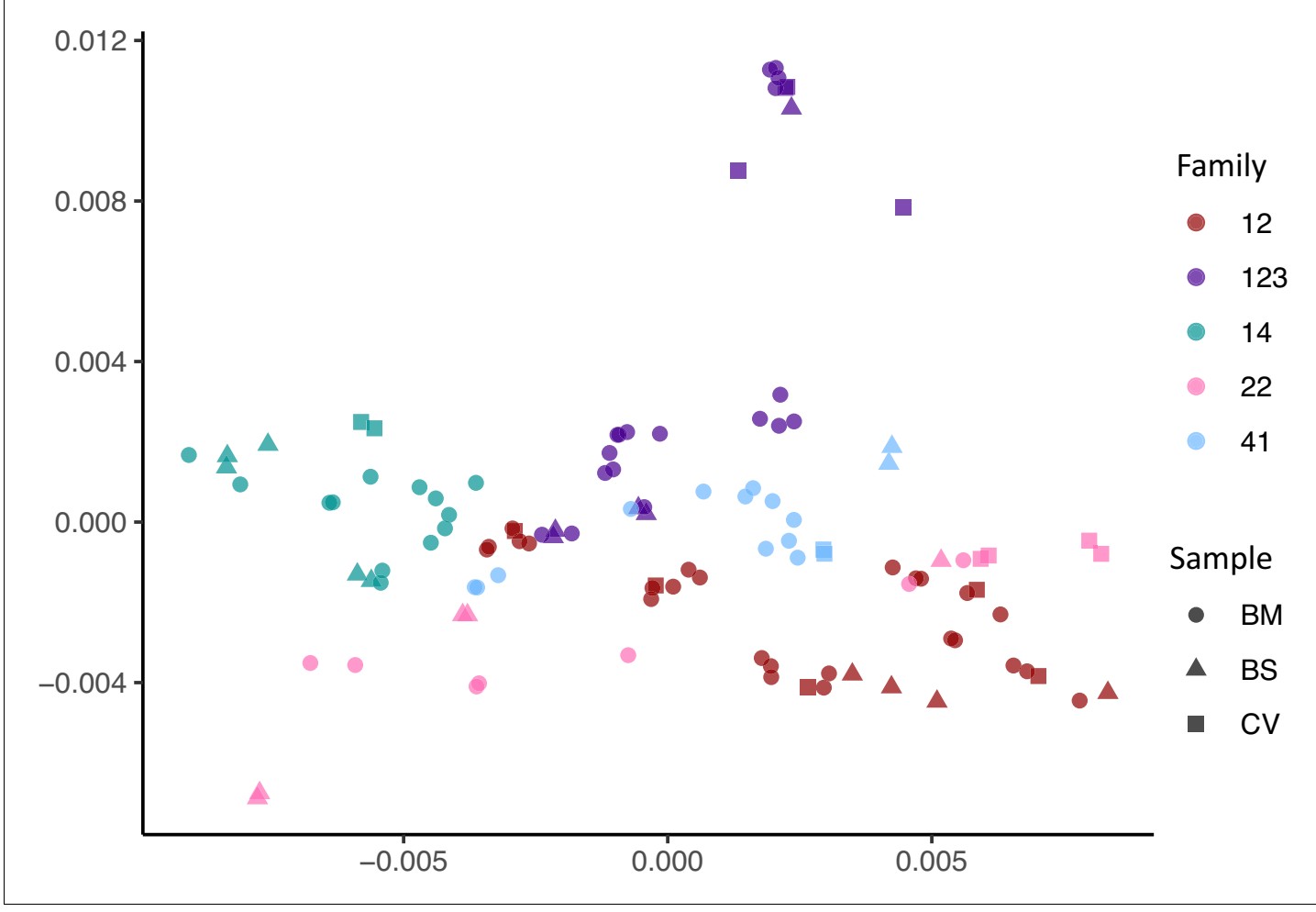

**Figure 3.** Multidimensional scaling showing clustering of haplotype sequences by family. Colours indicate the families; shapes indicate the types of sample.

The online version of this article includes the following figure supplement(s) for figure 3:

**Figure supplement 1.** Pairwise differences between haplotypes within a family.

**Figure supplement 2.** Maximum-likelihood phylogenetic tree to show haplotypes clusters (genotypes).

**Figure supplement 3.** Distribution of pairwise evolutionary distances for haplotypes within families.

tended to cluster by family group albeit with clear evidence of distinct clusters even within a family, for example family 22.

The presence of mixed infections within a single family was supported by data showing that a subset of the sequence haplotypes within each family had pairwise distances as great as those between unrelated GenBank sequences (*Figure 3—figure supplement 1*). Within-family phylogenetic analysis (*Figure 3—figure supplement 2*) shows distinct clusters of the phylogenetically related sequence haplotypes recovered from breast milk, cervix, and baby, likely to represent variants forming distinct viral strains (*Figure 3—figure supplement 2*). Based on the distribution of pairwise distances (see Materials and methods, *Figure 3—figure supplement 3*), we clustered similar haplotypes together into strains henceforth termed genotypes, so that all members of a cluster have a pairwise evolutionary distance with all other members less than 0.017, resulting in 26 clusters that we refer to as genotypes. In no cases did haplotypes from different families fulfil our clustering criterion confirming that haplotypes were not shared between unrelated families.

For ease of reference, genotypes were coloured differently, with the genotype predominating in the first cervical sample of each family coloured red (*Figure 3—figure supplement 2*). Other genotypes were coloured by their phylogenetic and pairwise distances from this genotype (*Figure 3—*

*figure supplement 2*). From our data, we identified at total of 26 genotypes with between 3 and 9 genotypes for each family (*Figure 3—figure supplement 2*).

To elucidate the relationship between maternal and infant genotypes, we plotted the abundance of each within a sample over time (*Figure 4*). All five mothers were infected with multiple genotypes in breast milk. In many cases, genotypes within a single maternal sample were as genetically distant as unrelated database sequences, suggesting the presence of multiple distinct CMV strains (*Figure 3—figure supplement 2*, *Figure 4*). Relative genotype abundances present in breast milk changed over time. One unique genotype appeared in the breast milk of mother 22 at 6 weeks, disappearing from a subsequent sample (*Figure 4*). This genotype was genetically distinct not only from other genotypes in family 22 but also from genotypes in all other families, reducing the likelihood that it was a contaminant and may therefore have represented a new reinfection or reactivation of pre-existing latent infection. All cervical samples showed a single dominant genotype (*Figure 4*), including mother 12, whose sample was more diverse and found to contain low levels of other genotypes. Overall, the data point to compartmentalization of CMV populations between cervix and breast milk.

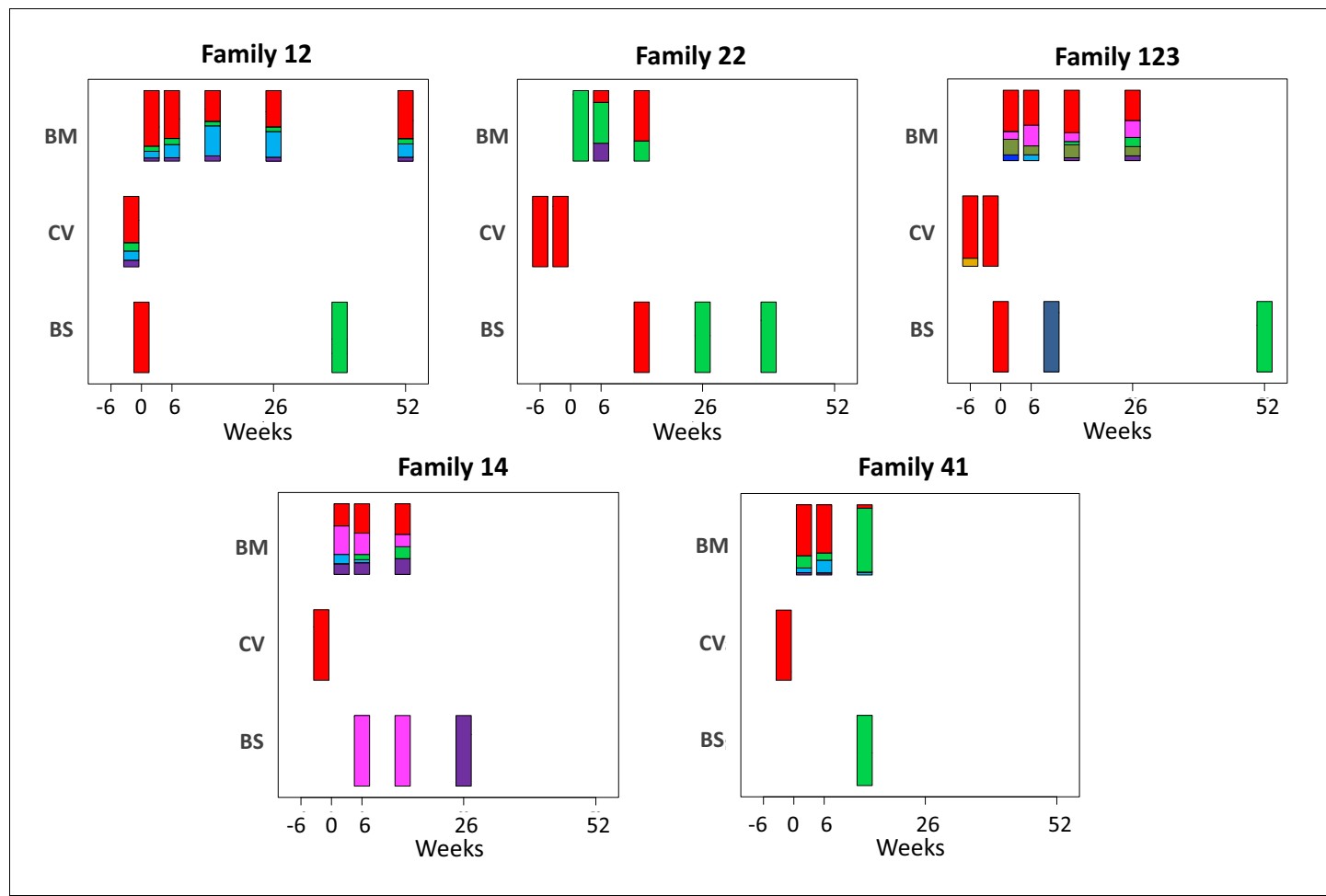

**Figure 4.** Abundance of haplotypes within each sample plotted for breast milk (BM), cervix (CV), and blood spots (BS). The timing of sampling is shown along the x axis. For ease of reference, the genotype containing the most abundant haplotype present in the cervix is coloured red for each family. Thereafter sequences that are genetically closest to the red genotype (*Figure 3—figure supplement 2*) are coloured magenta. Genotypes that are as distant from the cervical genotype as unrelated GenBank sequences are coloured shades of green, blue, and purple. Single variants are coloured in shades of the nearest genotype.

The online version of this article includes the following figure supplement(s) for figure 4:

**Figure supplement 1.** Boxplot showing number of haplotypes reconstructed in relation to read depth.

## Transmission bottlenecks

CMV genomes from individual infant blood spots also showed lower diversity (*Figure 2—figure supplement 1*), and predominance of one genotype (*Figure 4*), including samples with good sequence read depth, for example Baby12 DEL and 9M; Baby14 6W, 14W, and 6M; Baby22 14W; and Baby123 10W and 12M (*Table 1*), indicating the likelihood of a bottleneck in mother-to-child transmission. Two infants (families 12 and 123, *Figure 1*) who tested positive at birth were first infected with the genotype present in the greatest abundance in the cervix (*Figure 4* and *Figure 3—figure supplement 2*). The same pattern was found in a third infant (family 22) whose first sample at 2 weeks of age tested positive (*Figure 2*, *Figure 3—figure supplement 2*, and *Figure 4*). Interestingly, all three of these congenitally infected infants were subsequently re-infected with distinct genotypes present in breast milk (*Figure 4*). Two infants with initially two (family 14) and three (family 41) negative tests from birth onwards, first became positive at 6 and 10 weeks, respectively. The genotypes detected in the blood spots from both of these infants were present in breast milk and differed from the most abundant genotype in cervix (*Figure 4*).

## Subsampling to control for the impact of read depths

To determine the degree to which results were affected by the quality of sequence, we subsampled reads of different samples to show that sample diversity calculations are robust at read depths of $\geq 5$ (*Figure 2—figure supplement 2*); eight of the 18 blood spots and four of seven cervical samples had mean read depth of $\geq 10$ (*Table 1*) and all except one were of low diversity (*Figure 2—figure supplement 1*). To determine the extent to which read depth affected haplotype frequencies, the 12 month breast milk sample from mother 12, which had a mean read depth of 779.72 and five haplotypes (*Figure 3—figure supplement 2*), was subsampled down to mean read depth of <4 (*Figure 4—figure supplement 1*). All of the haplotypes in this sample were present for read depths of 22 or more, with three haplotypes identified even at the lowest read depth. Nine of 10 cervical and blood spot samples from four families with read depths of >20 (*Table 1*) had either single genotypes or multiple closely related variants (*Figure 4*), supporting previous conclusions around compartmentalization and transmission bottlenecks (*Renzette et al., 2011*).

## Genotype compartmentalization

Given the observation of multiple haplotypes in each of the mother–baby pairs, we can ask whether certain genotypes are more likely than others to be found in different compartments and whether there are common characteristics of the genotypes observed in similar compartments in different individuals. In order to address this question, we considered all possible subsets of between two and five genotypes where each genotype was derived from a different mother–baby pair. We then used fixation index ($F_{ST}$) to compare the genetic similarities of all of the genotypes in this set relative to the remaining genotypes. p-Values and false discovery rates (FDRs) for each pair were calculated using non-parametric bootstrapping. In order to compare various subsets, we computed a confidence-weighted sum of $F_{ST}$ (cws$F_{ST}$) values for each subset. The distribution of cws$F_{ST}$ values is shown in *Figure 5—figure supplement 1*. As can be seen, there are a large number of subsets with significant cws$F_{ST}$ values, far in excess of what is observed for scrambled sequences (black line).

The sum weighted $F_{ST}$ value for the subset of five genotypes that predominated in the cervical samples was not significantly different from other subsets, suggesting overall that genotypes that predominated in the cervix of these women were less closely related than most other comparisons (*Figure 5—figure supplement 1*, black arrow). Intriguingly, however, the subset of cervical genotypes from mother–baby pairs 12, 22, and 123 had a sum weighted $F_{ST}$ with a value greater than 99.6% of the other subsets (*Figure 5—figure supplement 1*, blue arrow), indicating a strong signal of inter-patient viral convergence. These genotypes were from the three mother–baby pairs with proven congenital infection based on first detection of CMV in the baby at $\leq 2$ weeks of age, and in whom the baby's genotype was identical to that predominating in cervix. In contrast, the predominant cervical genotypes from mothers 14 and 41 showed low levels of relatedness (*Figure 5—figure supplement 1*, red arrow). The infant strains from 14 and 41 were most closely related to those from their mothers' breast milk (*Figure 3—figure supplement 2* and *Figure 4*).

The $F_{ST}$ analysis identified 19 genes as likely to be contributing to the genetic similarity between congenitally transmitted genotypes from mothers 12, 22, and 123 (FDR < 0.05) (*Figure 5*). The

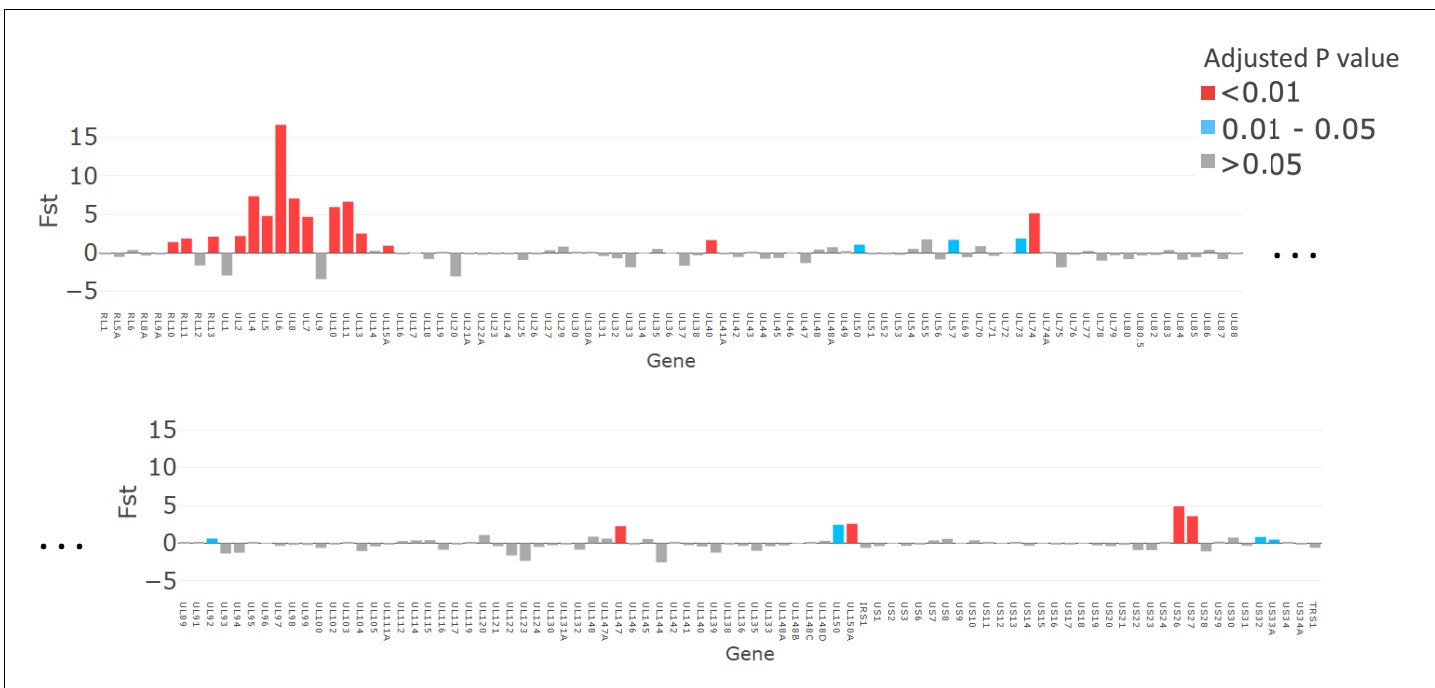

**Figure 5.** The magnitude of fixation index ($F_{ST}$) values plotted for each gene (x axis). p-Values, adjusted with false discovery rate, are shown in red for p<0.01, grey for p>0.05, and turquoise for p=0.01–0.05.

The online version of this article includes the following figure supplement(s) for figure 5:

**Figure supplement 1.** Distribution of confidence-weighted sums of $F_{ST}$ (cws$F_{ST}$) values for all subsets of two (cyan), three (purple), four (green), and five (magenta) genotypes from different mother–baby pairs.

**Figure supplement 2.** Heatmap showing genes identified as significant in $F_{ST}$ analysis are robust to changes in the number of clusters.

comparison between these congenitally transmitted and other genotypes generally yielded the same genes when the pairwise difference was varied to cluster haplotypes into more or fewer genotypes (*Figure 5—figure supplement 2*), suggesting that this finding is not an artefact of decisions about haplotype clustering.

## Discussion

We used next-generation sequencing and haplotype reconstruction of individual CMV genomes, obtained from samples of HIV-infected women and their infants, to identify mixed infections, compartmentalization, and distinct viral-genotype associations with transmission of CMV from mother-to-infant. Breast milk CMV showed high nucleotide diversity and, as has been previously reported (*Suárez et al., 2019*), contained a mixture of viral genotypes, some of which were as genetically distant from each other as unrelated GenBank sequences and can therefore be considered distinct viral strains. Cervical samples were of low nucleotide diversity and dominated by a single viral genotype that was, with one exception, present in lower abundance in breast milk. Our data fit with most but not all (*Puchhammer-Stöckl et al., 2006*) previous reports of CMV within-host compartmentalization based on genotyping of subgenomic fragments (*Hage et al., 2017*; *Kadambari et al., 2017*; *Ross et al., 2011*; *Renzette et al., 2013*). We found little evidence for widespread new superinfecting or reactivating viruses in these mothers. In line with the findings from the immunosuppressed RhCMV monkey model of congenital infection, cCMVi (*Vera Cruz et al., 2020*) genotypes (strains) comprised families of closely related haplotypes. However, unlike the finding for congenitally transmitted gB and gL RhCMV variants, even where we found transmission of one genotype, maternal and infant haplotypes were not completely identical either in early, potentially congenital CMV infections or in postnatally transmitted viruses from breast milk. Neither were haplotypes sampled at different times from maternal breast milk conserved, suggesting a

measure of de novo mutation in this patient group, in line with the previous findings (*Sackman et al., 2018*).

Our method of reconstructing viral haplotypes in serial samples provides insights into the natural history of CMV infection. While all mothers had mixtures of genotypes in breast milk, the proportions changed over time for some (families 22 and 41) and remained more stable in others. Whether expanding genotypes in mothers 22 and 41 had been recently acquired is not known but would be consistent with incident reinfection. In contrast, all infants were initially infected with a single genotype (*Figure 4*), supporting a bottleneck to CMV transmission (*Cudini et al., 2019*; *Vera Cruz et al., 2020*; *Stanton et al., 2010*). Apparent reinfection by viruses present in breast milk occurred in all four infants with multiple samples (*Figure 4*). We posit that the appearance of a new strain in an infant sampled from birth can confidently be interpreted as a newly acquired exogenous virus rather than reactivation of a previously undetected one. In all cases, the reinfecting strains were genetically distant from and replaced the previously dominant strain (*Figure 4*). Taken together with the rise and fall of infant CMV viral loads over time (*Figure 1*), this pattern is consistent with immunity against the infants' first CMV strain not being protective against reinfection with antigenically distinct strains, a concept that can be further tested. Of note, reinfection with the closely related strains also appears to occur readily with both human CMV and animal models (*Boucoiran et al., 2018*; *Hansen et al., 2010*). Repeated reinfection with distinct strains may explain the high genetic variability observed between sequential samples in early sequencing studies of CMV genomes from congenitally infected infants (*Pokalyuk et al., 2017*; *Renzette et al., 2013*).

Those infants who tested positive at <3 weeks from birth were congenitally infected by definition (*Boppana et al., 1999*). In contrast, we cannot formally rule out cCMVi in the two others who were classified as having postnatal infection, since sensitivity of PCR detection of CMV DNA in newborn blood spots is only approximately 84% (*Wang et al., 2015*), and newborn saliva or urine were not available. However, this is unlikely given that only a small minority of infants have cCMVi, even among those born to HIV-infected women. Furthermore, it is striking that genotypes in babies with proven cCMVi were highly similar to maternal cervical genotypes, while those with negative tests for the first 6 weeks of life were not, and the strains detected later in the blood of these two infants were most similar to those in their mothers' breast milk.

While it has previously been noted that a severe genetic bottleneck occurs during CMV transmission from mother to foetus or infant (*Sackman et al., 2018*; *Renzette et al., 2013*; *Mayer et al., 2017*), it remains unknown whether CMV transmitted/founder virus populations share genotypic features that confer a fitness advantage for establishing an initial infection, such as seen in HIV (*Joseph et al., 2015*). Notwithstanding the apparent dominance of one genotype in each of the cervical samples, our analysis did not show evidence for inter-patient convergence of cervical genotypes per se. Rather the three cervical genotypes that were detected in babies 12, 22, and 123, who were infected at birth showed a higher level of genetic similarity than over 99.6% of other subset comparisons and much greater than would be expected by chance (black line) (*Figure 5—figure supplement 1*). Nineteen genes (*Figure 5*, *Table 2*) had particularly high (p<0.01) similarity scores. Twelve of the 19 genes with the highest similarity scores (*Figure 5*) are part of the highly diverse RL11 gene family. Uniquely, RL11 genes form an island of linkage within the otherwise highly recombinant CMV genome (*Lassalle et al., 2016*). Phylogeny of primate CMV RL11 complexes recapitulates the evolutionary history of the cognate host, suggesting it to be a potential driver of CMV co-evolution and speciation (*Lassalle et al., 2016*). It is intriguing that RL11 family proteins influence tissue tropism (*Stanton et al., 2010*) or are immunomodulatory (*Stanton et al., 2010*; *Cortese et al., 2012*; *Van Damme and Van Loock, 2014*; *Pérez-Carmona et al., 2018*; *Bruno et al., 2016*; *Gabaev et al., 2014*). Together with its functional properties (*Table 2*) and extreme diversity (*Lassalle et al., 2016*), the possibility that within-species CMV RL11 gene-family variation may also influence within-host viral adaption to different compartments and/or transplacental transmission presents a tractable hypothesis that can now be tested. cCMVi is thought to occur primarily through maternal viremia followed by replication in placental cytotrophoblasts resulting in spread to the foetus (*Pereira et al., 2017*). The three mothers who transmitted their viruses congenitally had higher cervical viral loads than mothers whose babies become infected postpartum (*Figure 1*). Analysis of data from the whole cohort of mothers confirmed that women who transmitted CMV in utero had mean cervical CMV vial loads at 38 weeks that were 0.83 $\log_{10}$ copies/ml (SD = 1.0, p=0.02) higher than women who did not transmit CMV in utero (data not shown) (*Roxby et al., 2014*). We therefore

**Table 2.** Open reading frames (ORFs) identified by fixation index ($F_{ST}$) as being significantly more similar in strains transmitted prenatally.

LD: Found to contain one of 33 hotspots of genetic linkage disequilibrium (*Lassalle et al., 2016*).

| ORF | LD | Function |
|---|---|---|
| UL10 | Y | Putative membrane glycoprotein. Immunosuppressive, Impairs T cell function (*Bruno et al., 2016*) |
| UL11 | Y | Membrane glycoprotein. Modulates T cell signalling/function (*Gabaev et al., 2014*; *Arcangeletti et al., 2015*) |
| UL13 | | Unknown function |
| UL4 | Y | Putative membrane glycoprotein (*Van Damme and Van Loock, 2014*) |
| UL5 | | Putative membrane glycoprotein (*Van Damme and Van Loock, 2014*) |
| UL6 | Y | Putative membrane glycoprotein (*Van Damme and Van Loock, 2014*) |
| UL7 | Y | Membrane glycoprotein. Modulates chemo- and/or cytokine-signalling function (*Pérez-Carmona et al., 2018*) |
| UL8 | Y | Transmembrane glycoprotein. Inhibits proinflammatory cytokines (*Pérez-Carmona et al., 2018*) |
| US26 | | Unknown function |
| US27 | Y | Membrane glycoprotein. Activates CXCR4 signalling to increase human cytomegalovirus replication (*Frank et al., 2019*) |
| UL150A | | Fibroblast and epithelial cell entry (*Houldcroft et al., 2016*) |
| UL2 | | Putative membrane glycoprotein (*Van Damme and Van Loock, 2014*) |
| RL11 | Y | Membrane glycoprotein. Binds IgG Fc domain involved in immune regulation (*Van Damme and Van Loock, 2014*) |
| UL147 | | α-Chemokine homologue (*Katoh and Standley, 2013*; *Paradis and Schliep, 2019*) |
| UL40 | | Control of NK recognition (*Heatley et al., 2013*) |
| RL13 | Y | Glycoprotein, repression of replication, binds IgG domain immune regulation (*Stanton et al., 2010*; *Cortese et al., 2012*) |
| RL10 | | Membrane glycoprotein |
| UL57 | | Ss DNA binding protein (*Van Damme and Van Loock, 2014*) |
| UL50 | | Nuclear Egress complex. Reduces interferon-mediated antiviral effect (*DeRussy et al., 2016*) |

speculate that virus sampled in the cervix is representative of CMV populations that infect and cross the placenta and that a possible explanation for our findings is that the properties that promote replication to higher titres in genital tissue may also predispose to transplacental infection.

Other genes with high similarity ($F_{ST}$) scores include US27, which codes for a G-protein-coupled receptor homologue that modulates signalling of the CXCR4 chemokine and may have a role during viral entry and egress (*Frank et al., 2019*), and US26 whose function is unknown. Less marked but still significantly different from non-congenitally transmitted strains, UL40 protein (*Heatley et al., 2013*) modulates natural killer (NK) cell function. NK cells are the most abundant lymphocytes in placental tissue (*Pereira et al., 2017*), while UL50 is also immunomodulatory (*Lee et al., 2018*; *DeRussy et al., 2016*). Finally, UL74, coding for glycoprotein O, which is highly significantly similar in all bar one comparisons (*Figure 5—figure supplement 2*), is part of the glycoprotein complex that is critical for tropism and entry into both fibroblasts and epithelial cells (*Wu et al., 2017*). Of interest, gB and gL, which showed considerable diversity in the congenital RhCMV model, were, as might be expected, not represented among the genes sharing significant genetic similarity in our analysis. One possibility that would unite our findings and those of the congenital RhCMV model is that CMV transmission bottlenecks are agnostic of variation in genes not implicated in transmission.

Being born to HIV-infected women is a major risk factor for cCMVi as well as long-term CMV-related complications, whether or not the child acquires HIV (*Garcia-Knight et al., 2017*; *Gompels et al., 2012*). We show here that, irrespective of the route of first infection, HEU children frequently acquire repeated infections with different CMV viruses within the first year of life. Preliminary evidence suggests that breast milk of HIV-uninfected women may have lower CMV viral loads and carry fewer strains (*Arcangeletti et al., 2015*). If this is true, the possibility that HEU, as well as HIV-infected, infants are exposed to greater numbers of CMV strains during infancy when compared with HIV-uninfected infants may provide an explanation for their worse clinical outcomes, a hypothesis that can now be tested in prospective studies. Similarly, these methods promise to be invaluable

for studying the role of maternal CMV reinfection during pregnancy, a question of central importance in the field (*Britt, 2017*).

This study potentially provides several new insights into the pathogenesis of CMV infection. However, the study is limited by the small number of subjects, the fact that all women were HIV-1 infected and the lack of samples and data to absolutely confirm the route of CMV acquisition by these infants. Because we were only able to analyse maternal breast milk, cervical samples, and infant blood, and only intermittently, it is possible that some transmitted viral variants were not captured. Some, particularly cervical and blood spot samples, had low CMV viral loads and, as a result, suboptimal genome coverage. Mapping data confirmed that in these cases, sequence loss was random, excluding the possibility of systematic bias. To further address this potential bias, we subsampled samples with good coverage to identify read-depth thresholds above which the diversity estimation is robust and haplotype frequency to 5% and above is preserved (*Figure 2—figure supplement 2* and *Figure 4—figure supplement 1*). Analysis of only those samples with read depths above the identified thresholds supported our overall conclusions. The quality of the sequence and the numbers of samples allowed for conclusions to be drawn at gene level only and precluded robust identification of putative motifs or single-nucleotide polymorphisms associated with biological differences.

In summary, by reconstructing the individual CMV haplotypes, we found evidence for mixed CMV infection in HIV-infected women, and compartmentalization of viral strains between cervical and breast milk. Infants appeared usually to acquire one virus genotype initially, indicating a transmission bottleneck, though subsequent reinfection with a second virus from maternal breast milk was common. We also found that viruses transmitted congenitally resembled the virus genotypes that were present at highest abundance in cervix, and shared genetic features that distinguished them from CMV strains predominating in breast milk and in the cervices of women whose infants were apparently first infected post-partum. These data provide new testable insights into the pathogenesis of CMV transmission from mothers to their infants, as well as tools to unravel the importance of viral diversity for reinfection and congenital transmission, questions that are central to the development of a vaccine to prevent the global burden of disease due to CMV.

## Materials and methods

Samples were approved for research by the Institutional Review Board of the University of Washington and the Ethics and Research Committee of Kenyatta National Hospital IRB NCT00530777 and sequenced under the ULCP Biobank REC approval. Approval for use of anonymized residual diagnostic specimens was obtained through the University College London/University College London Hospitals (UCL/UCLH) Pathogen Biobank National Research Ethics Service Committee London Fulham (Research Ethics Committee reference: 12/LO/1089). Informed patient consent was not required.

### Patient specimens

Mother–child pairs were selected from a randomized, placebo-controlled trial to determine the impact of twice-daily valacyclovir (500 mg) on breast milk HIV RNA viral load in HIV-1/HSV-2 co-infected women (NCT 00530777). Trial design, participant characteristics, and follow-up have been reported elsewhere (*Drake et al., 2012*; *Roxby et al., 2014*; *Slyker et al., 2014*) and the University of Washington Institutional Review Board and Kenyatta National Hospital Research and Ethics Committee approved the research. Women received short course antiretrovirals for prevention of mother-to-child HIV transmission, but no women or infants received combination antiretroviral therapy, as the study was conducted before recommendations for universal treatment. All women were HIV-1, HSV-2, and CMV co-infected. For this CMV genomics study, we selected five mother–infant pairs from the placebo arm, who had well-defined timing of infant CMV infection. All infants were HIV exposed, and one was HIV infected. Women had cervical swabs and blood specimens collected at 34 and 38 weeks gestation. Maternal blood and infant dried blood spots were collected delivery, then postpartum at 2, 6, 10, 14, 24, 36, and 52 weeks. Breast milk was collected at all times after delivery. Blood plasma, cervical swabs, and breast milk supernatant (whey) were cryopreserved at –80℃ for the study of HIV and other co-infections.

## DNA extraction and CMV DNA measurement

Viral nucleic acids were extracted from blood plasma, dried blood spots, breast milk supernatant, and cervical swabs as previously described using the Qiagen UltraSens Viral Nucleic Acid extraction kit (*Roxby et al., 2014*). Quantitative real-time PCR was used to measure CMV DNA levels in these specimens (*Roxby et al., 2014*).

## Sure-select sequencing

Hybridization and library preparation were performed as previously described (*Houldcroft et al., 2016*). Briefly, extracted DNA was sheared by acoustic sonication (Covaris e220, Covaris Inc). DNA fragments underwent end-repair, A'-tailing, and (Illumina) adaptor ligation. DNA libraries were hybridized with biotinylated 120-mer custom RNA baits designed using all available CMV full genomes in GenBank for 16–24 hr at 65°C and subsequently bound to MyOne Streptavidin T1 Dyna-beads (ThermoFisher Scientific). Following washing, libraries were amplified (18 cycles) to generate sufficient input material for Illumina sequencing. Paired-end sequencing was performed on an Illumina MiSeq using the 500 cycle v2 Reagent Kit (Illumina, MS-102–2003). Samples were sequenced in four different batches by family group.

Reads generated were quality checked and mapped to the Merlin reference sequence followed by removal of duplicates using the CLC Genomics Workbench ver. 10.1. Consensus sequence was extracted with a minimum coverage of 2×. All consensus sequences along with other GenBank reference sequences were aligned using MAFFT 7.212 (*Katoh and Standley, 2013*) and refined by manual editing.

## Clustering

Pairwise distances between sequences were calculated using the dist.dna function from R package Ape v.5.3 (*Paradis and Schliep, 2019*). Sequences were clustered using multidimensional scaling as implemented by the cmdscale function from R package Stats v.3.6 (*Team, 2012*).

## Nucleotide diversity

Nucleotide diversity was calculated by fitting the observed variant frequency spectrum to the mixture of two distributions, one representing sequencing errors (represented by a Beta distribution) and the other representing true diversity (represented by a four-dimensional Dirichlet distribution plus delta function, the latter representing invariant sites). The parameters for these two distributions were optimized by maximizing the log likelihood. This framework allows all of the sequencing data to be used and does not require pre-filtering the data to remove sites with low read depth or few variants resulting in the favourable robustness to read depth, as shown in *Figure 2—figure supplement 2*. Software is available for download at GitHub Repository, https://github.com/ucl-pathgenomics/NucleotideDiversity (copy archived at swh:1:rev:20814eda934c539608b30e8fe21ead282046fa8b; *Pang et al., 2020b*).

## Haplotype reconstruction

Haplotype reconstruction was accomplished using HaROLD with default settings (*Pang et al., 2020a*). Details of this procedure are described in the associated publications. In brief, HaROLD employs a two-step process. The first step is based on the assumption that there are a limited number of haplotypes that are the same for all of the samples from a given mother/child data set, so that the differences in the frequencies of polymorphisms represent different mixtures of these haplotypes. HaROLD creates a set of haplotypes from each data set by selecting the set of haplotypes whose linear combinations optimally accounts for the observed variant frequencies. The number of haplotypes is chosen to maximize the log likelihood of the observed frequencies. The second step involves relaxing the assumption of constant haplotypes, with each sample treated individually. For each sample, reads are assigned probabilistically to the various haplotypes generated by the first step. These haplotype sequences and frequencies are then adjusted based on the assigned reads. The reads are then re-assigned to these adjusted haplotypes, and the procedure is repeated until convergence. During this process, haplotypes can be merged if that decreases the Akaike information criterion (*Akaike, 1973*). This procedure results in a set of haplotypes for each sample, loosely based on the haplotypes derived from the first step.

## Haplotype trees

Maximum-likelihood trees of the haplotypes from each family were computed using RaxML v8.2.10, implementing the GTR model, with 1000 bootstrap replicates (*Stamatakis, 2014*).

## Haplotype clustering

The haplotypes for each mother/baby data set were divided into genotypes. We calculated the pairwise evolutionary distance (the sum of distances on the evolutionary tree between the haplotypes and their latest common ancestor) for all pairs of haplotypes in each family. As shown in *Figure 3— figure supplement 3*, the observed distribution of such pairwise distances fits the sum of a gamma distribution (69.3%, alpha = 19.5, beta = 0.0015) and an exponential distribution (30.7%, mean = 0.01), indicative of two classes of relationships – pairs of sequences that are highly similar, modelled by the exponential, representing small accumulated variations, and pairs that are more distinct, represented by the gamma distribution. We chose the crossing point of these two distributions, at a cut-off distance of 0.017, as differentiating small variations from larger differences (*Figure 3—figure supplement 3*). We then grouped the haplotypes into clusters so that all members of a cluster have a pairwise evolutionary distance with all other members less than 0.017, resulting in 26 clusters that we refer to as genotypes. We used these groups to assign colours to the different haplotype clusters (genotypes) in *Figure 4* and *Figure 3—figure supplement 2*.

We used $F_{ST}$ to identify sequence characteristics associated with sets of genotypes. Consensus sequences were constructed for each genotype. $F_{ST}$ values, representing the genetic difference between a subset of genotypes and the other genotypes, were calculated for each gene. p-Values and corresponding FDRs were estimated by non-parametric bootstrapping, through scrambling the bases at each position amongst the clusters. The results are shown for the 26 genotypes obtained with a cut-off distance of 0.017; changing this cut-off resulted in increased or decreased numbers of genotypes, but yielded similar results, especially for the more confident identifications (*Figure 5— figure supplement 2*).

## Evaluating the similarity between subsets of genotypes

We use $F_{ST}$ values to identify similarities between individual genes from subsets of genotypes compared with the other genotypes. In order to compare the magnitude of the similarities of different subsets, we would like to take the sum of the $F_{ST}$ values for all genes where the similarities are real and not the result of random associations. As we cannot definitively identify these genes, we instead consider the sum of the $F_{ST}$ values for all genes weighted by our confidence that the $F_{ST}$ value is significant, represented as one minus the FDR.

## Data availability

Sequence reads have been deposited in NCBI Sequence Read Archive under BioProject ID PRJNA605798.

All software used are available for download at GitHub Repository, https://github.com/ucl-pathgenomics/NucleotideDiversity and https://github.com/ucl-pathgenomics/HAROLD.

## Acknowledgements

We acknowledge the support of the MRC/NIHR UCLH/UCL Biomedical Research Centre funded Pathogen Genomics Unit. This work was funded by EUFP7 grant 304875 (PI Breuer), Wellcome Trust grant 204870 (PI Griffiths), NIH National Institute of Allergy and Infectious Diseases grant AI087369 (PI Slyker), AI027757 (PI Slyker, Holmes), AI076105 and K24 AI087399 (Farquhar), National Institute of Child Health and Human Development HD057773–01, HD054314 (Farquhar). JP is funded by a Rosetrees Trust PhD Studentship M876. SM and J Bryant are funded by Henry Wellcome fellowships. J Breuer receives funding from the UCL/UCLH NIHR Biomedical Research Centre.

## Additional information

### Funding

| Funder | Grant reference number | Author |
|---|---|---|
| EUFP7 | 304875 | Judith Breuer |
| Wellcome Trust | 204870 | Paul Griffiths |
| National Institute of Allergy and Infectious Diseases | AI087369 | Jennifer A Slyker |
| National Institute of Allergy and Infectious Diseases | AI027757 | Jennifer A Slyker |
| National Institute of Allergy and Infectious Diseases | AI076105 | Carey Farquhar |
| National Institute of Allergy and Infectious Diseases | AI087399 | Carey Farquhar |
| National Institute of Child Health and Human Development | HD057773-01 | Carey Farquhar |
| National Institute of Child Health and Human Development | HD054314 | Carey Farquhar |
| Rosetreees Trust PhD Studentship | M876 | Juanita Pang |
| UCL/UCLH NIHR Biomedical Research Centre | | Judith Breuer |
| Sir Henry Wellcome Fellowship | | Sofia Morfopoulou |
| Sir Henry Wellcome Fellowships | | Josephine Bryant |

The funders had no role in study design, data collection and interpretation, or the decision to submit the work for publication.

### Author contributions

Juanita Pang, Data curation, Software, Formal analysis, Validation, Investigation, Visualization, Methodology, Project administration, Writing - review and editing; Jennifer A Slyker, Resources, Data curation, Funding acquisition, Investigation, Writing - review and editing; Sunando Roy, Josephine Bryant, Juliana Cudini, Data curation, Formal analysis, Investigation, Writing - review and editing; Claire Atkinson, Alison C Roxby, Data curation, Investigation, Writing - review and editing; Carey Farquhar, Conceptualization, Data curation, Funding acquisition, Investigation, Methodology, Writing - review and editing; Paul Griffiths, Funding acquisition, Investigation, Methodology, Writing - review and editing; James Kiarie, Conceptualization, Investigation, Writing - review and editing; Sofia Morfopoulou, Data curation, Formal analysis, Validation, Investigation, Visualization, Methodology, Writing - review and editing; Helena Tutil, Data curation, Formal analysis, Methodology, Writing - review and editing; Rachel Williams, Data curation, Formal analysis, Writing - review and editing; Soren Gantt, Conceptualization, Validation, Investigation, Methodology, Writing - review and editing; Richard A Goldstein, Conceptualization, Resources, Data curation, Software, Formal analysis, Supervision, Validation, Investigation, Visualization, Methodology, Writing - original draft, Project administration, Writing - review and editing; Judith Breuer, Conceptualization, Formal analysis, Supervision, Funding acquisition, Investigation, Methodology, Writing - original draft, Project administration, Writing - review and editing

### Author ORCIDs

Juanita Pang (ID) https://orcid.org/0000-0002-4792-5903
Soren Gantt (ID) http://orcid.org/0000-0001-5743-3606

Richard A Goldstein (iD) https://orcid.org/0000-0001-5148-4672
Judith Breuer (iD) https://orcid.org/0000-0001-8246-0534

**Decision letter and Author response**
Decision letter https://doi.org/10.7554/eLife.63199.sa1
Author response https://doi.org/10.7554/eLife.63199.sa2

## Additional files

**Supplementary files**
• Transparent reporting form

**Data availability**
Sequence reads have been deposited in NCBI Sequence Read Archive under BioProject ID PRJNA605798.

The following dataset was generated:

| Author(s) | Year | Dataset title | Dataset URL | Database and Identifier |
|---|---|---|---|---|
| Pang J, Slyker JA, Roy S, Bryant J, Atkinson C, Cudini J, Farquhar C, Griffiths P, Kiarie J, Morfopoulou S, Roxby AC, Tutil H, Williams R, Gantt S, Goldstein RA, Breuer J | 2020 | Cytomegalovirus whole genome sequencing | https://www.ncbi.nlm.nih.gov/bioproject/PRJNA605798 | NCBI BioProject, PRJNA605798 |

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
