## [Decision Letter]

**Acceptance summary:**

This paper reports studies using genome sequencing and computational tools that allow haplotype reconstruction to follow individual cytomegalovirus genomes (CMV) within mixed infections in transmission by HIV-positive mothers to their infants. The report shows how novel genomic approaches and computational tools can be used to gain insights into the biology of congenital and postnatal CMV transmission. The reported observations have major implications for the understanding of the viral genetic correlates of tissue tropism and other biological properties, as well as for the development of vaccines to prevent congenital infection.

**Decision letter after peer review:**

Thank you for submitting your article "Mixed CMV genotypes in HIV positive mothers show compartmentalization and distinct patterns of transmission to infants" for consideration by *eLife*. Your article has been reviewed by three peer reviewers, and the evaluation has been overseen by a Reviewing Editor and Anna Akhmanova as the Senior Editor. The following individual involved in review of your submission has agreed to reveal their identity: Nanda Ramchandar (Reviewer #2).

The reviewers have discussed the reviews with one another and the Reviewing Editor has drafted this decision to help you prepare a revised submission.

Summary:

This is an interesting study that investigates the role of CMV (cytomegalvirus) genotype in congenital and post-natal transmission of CMV to infants born to HIV-positive mothers. The authors describe in a small group of subjects evidence of infection involving multiple strains that cluster by sample type and by family. By sampling at least 3 different sites (Cervix, Breast milk and infant blood) they trace the course of congenital CMV transmission and infection. Advances in high throughput sequencing (HTS) and HTS analysis allow the resolution of CMV haplotypes and therefore the subsequent tracking of CMV genomes from mixed CMV populations. The data presented build upon and add to the literature on this important issue and contribute to our understanding of the underlying pathophysiology of CMV transmission from infected mothers to their infants.

Essential revisions:

Reviewer 1:

1) The authors state that cervical samples were of low nucleotide diversity and dominated by a single viral genotype. How can the authors be sure that this is not due to the clearly lower viral load in these specimens?

2) Has the coverage of all CMV genes been sufficient and at a degree, which does justify conclusions withdrawn from Figure 5?

3) In the Materials and methods it is stated that 5 non-HIV infected children were selected, while according to the results obviously one HIV-infected child was included.

Reviewer 2:

1) The Keys for the Figure 2 and Figure 2—figure supplement 1 MDS plots needs to be clarified (appears to be missing the colors and shapes). I imagine that there was clustering by both family and sample type.

2) What was the HIV viral load in the mothers? What was the HIV viral load in Infant 12? Why was maternal blood not available?

Reviewer 3:

1) Introduction: It would be helpful to have a brief breakdown of the geographical distribution for their figures for infection?

2) “Due to their abundance in the community” – I wasn't clear what “their abundance” was referring to here?

3) “Cervical, breast milk, and blood viral loads, and time of sample collection for the five mother infant pairs studied are shown in Figure 1.” I can only see infant blood – not maternal blood. If correct then the above should be changed to infant blood?

4) Figure 2—figure supplement 1. Unclear how to interpret this figure. No indication what samples the colours and symbols refer to – please correct. If the symbols are the same as shown for Figure 1, is breast milk really more diverse than infant blood and cervix – looks like there might just be more samples. Would benefit from statistical analysis.

5) The authors show evidence of what they refer to as “inter-patient” viral convergence. They need to explain much more clearly what is meant by this as it seems critical.

6) I didn't understand Figure 5 (nor were the x-axis labels legible) or Figure 5—figure supplement 2.

Nor did I understand the paragraph:

“The FST analysis identified 19 genes as likely to be contributing to the genetic similarity between congenitally transmitted genotypes from mothers 12, 22, 123 (FDR < 0.05) (Figure 5). The comparison between these congenitally-transmitted and other genotypes generally yielded the same genes when the pairwise difference was varied to cluster haplotypes into more or fewer genotypes (Figure 5—figure supplement 2), suggesting that this finding is not an artefact of decisions about haplotype clustering.”

“Rather the three cervical genotypes that were detected in babies 12, 22 and 123, who were infected at birth showed a higher level of genetic similarity than over 99.6% of other subset comparisons and much greater than would be expected by chance (black line) (Figure 5—figure supplement 1)” – so is this the explanation why they think there is a genetic bias towards the dominant virus that can get through the transit bottleneck – what might contribute to this bottleneck? Needs further explanation.

---

## [Author Response]

Essential revisions:Reviewer 1:1) The authors state that cervical samples were of low nucleotide diversity and dominated by a single viral genotype. How can the authors be sure that this is not due to the clearly lower viral load in these specimens?

We thank the reviewer for highlighting this important point. We also identified that samples with lower viral load generally have sequences with lower read depth (Figure 1—figure supplement 1) and that this could potentially interfere with identification of mixed infections. To address this we carried out several analyses. First, we confirmed that lower input viral loads result in low read depths (Figure 1—figure supplement 1). To determine the degree to which estimated nucleotide diversity was affected by lower read depth (i.e. viral load), we created samples with lower read depths by subsampling reads from samples with high read depths, allowing us to see how reducing the read depth affected the estimated diversity. As shown in Figure 2—figure supplement 2, sample diversity calculations are robust at average read depths of ≥5. Moreover, eight of the 18 blood spots and four of seven cervical samples had mean read depth ≥10, while family 12: cervix, baby delivery, family 22: baby 14 weeks family 14, baby 14 weeks and family 123: cervix 34 weeks, baby 10 weeks, and baby 12 months had read depths >30 (Table 1). All except family 12: cervix were of low diversity (Figure 2—figure supplement 1).

Finally, to determine whether low read depths (viral loads) interfere with identification of haplotypes, we subsampled reads from the 12-month breastmilk sample from mother 12, which had a mean read depth of 779.72 and five haplotypes (Figure 3—figure supplement 2). All of the haplotypes in this sample were present at a mean read depth of 22 or more, with loss of one haplotype, present at around 5% at read depths below 22 and loss of a second haplotype present at around 8% below read depths of 11 (Figure 4—figure supplement 1).

Thus we are confident that the low nucleotide diversity we observed is not due to the lower viral load and read depth of these samples. Moreover, although there is some fall off of low level haplotypes (~5% abundance) below read depths of around 20, the method is still able to identify higher abundance haplotypes even at very low read depths.

2) Has the coverage of all CMV genes been sufficient and at a degree, which does justify conclusions withdrawn from Figure 5?

We used haplotype sequences for the FST analysis shown in Figure 5. As shown in Figure 4—figure supplement 1, the program we used for haplotype reconstruction (HaROLD) performs well even at low read depth. Moreover, HaROLD’s approach is to consider all reads at each position, integrating data from every sample in which the haplotype is found and give a confidence level on how likely the base is genuine at that position. If the read coverage is not sufficient, the position will be indicated as a gap (missing data) and this position is excluded in all downstream analyses. In the FST test, lack of data simply gives no statistical support to that gene, meaning that genes with low coverage would not result in false positives. Undoubtedly there might be a small number of genes that we have missed (false negatives); however, by reconstructing haplotypes using longitudinally taken samples and creating a consensus, the numbers of gaps are minimised and thus the number of missing significant genetic associations reduced.

3) In the Materials and methods it is stated that 5 non-HIV infected children were selected, while according to the results obviously one HIV-infected child was included.

We apologise for the confusion. Infant 12 was HIV-infected. We have now corrected it in the Materials and methods section.

Reviewer 2:1) The Keys for the Figure 2 and Figure 2—figure supplement 1 MDS plots needs to be clarified (appears to be missing the colors and shapes). I imagine that there was clustering by both family and sample type.

The existing key shows the family by colour and sample type by shape. Clustering by family is observed, and the first sample from each infant clustered most closely with that of its mother.

2) What was the HIV viral load in the mothers? What was the HIV viral load in Infant 12? Why was maternal blood not available?

We have added the mothers’ blood plasma HIV viral load in Figure 1.

The Infant 12 blood sample was collected on dried blood spot for HIV PCR, which is the standard sample used for HIV diagnosis in infants. Unfortunately, no other blood was stored for the infants, and HIV viral loads were not estimated during the study.

Maternal blood samples from the first visit were available, and we estimated plasma CMV viral loads. However, the plasma CMV viral loads for all patients were below the limit of detection and thus whole genome sequencing was not possible.

Reviewer 3:1) Introduction: It would be helpful to have a brief breakdown of the geographical distribution for their figures for infection?

We have added the CMV prevalence rate in Kenyan pregnant women into the Introduction.

2) “Due to their abundance in the community” – I wasn't clear what “their abundance” was referring to here?

This sentence has been rewritten.

“Over two-thirds of infants with cCMVi are born to seropositive women, which constitute 88.4% of women in the Kenyan community from whom these study participants were drawn.”

3) “Cervical, breast milk, and blood viral loads, and time of sample collection for the five mother infant pairs studied are shown in Figure 1.” I can only see infant blood – not maternal blood. If correct then the above should be changed to infant blood?

Only infant blood was available in this study. This is now clarified in the paragraph. “Cervical, breast milk, and infant blood viral loads, and time of sample collection for the five mother-infant pairs studied are shown in Figure 1.”

4) Figure 2—figure supplement 1. Unclear how to interpret this figure. No indication what samples the colours and symbols refer to – please correct. If the symbols are the same as shown for Figure 1, is breast milk really more diverse than infant blood and cervix – looks like there might just be more samples. Would benefit from statistical analysis.

The existing key shows the family by colour and sample type by shape.

“It has previously been reported that a nucleotide diversity of 0.005 or above is likely to indicate a mixed infection [Cudini et al., 2019].” Only 1 out of 7 cervix sample and none of the 12 baby blood spot samples had a nucleotide diversity of > 0.005, while 16 out of 18 breastmilk samples had a nucleotide diversity above 0.005. Mann-Whitney test gives a p value of 1.619e-07 and 9.69e-6 respectively for comparing cervix and baby blood spot to breastmilk samples, showing that breastmilk samples have significantly higher nucleotide diversity.

5) The authors show evidence of what they refer to as “inter-patient” viral convergence. They need to explain much more clearly what is meant by this as it seems critical.

We have rewritten this section to clarify this point. We are interested in understanding whether there are genetic similarities between the genotypes from different mother-baby pairs found in similar compartments. In our analysis, we observed such similarities between genotypes that underwent congenital transmission. Assuming that the distribution in genotypes occurred after infection of the mother, this indicates that similar genetic changes occurred in multiple mothers and that the viruses with these genetic changes were more likely to transmit congenitally. This is what we mean by “viral convergence”.

6) I didn't understand Figure 5 (nor were the x-axis labels legible) or Figure 5—figure supplement 2.Nor did I understand the paragraph:“The FST analysis identified 19 genes as likely to be contributing to the genetic similarity between congenitally transmitted genotypes from mothers 12, 22, 123 (FDR < 0.05) (Figure 5). The comparison between these congenitally-transmitted and other genotypes generally yielded the same genes when the pairwise difference was varied to cluster haplotypes into more or fewer genotypes (Figure 5—figure supplement 2), suggesting that this finding is not an artefact of decisions about haplotype clustering.”“Rather the three cervical genotypes that were detected in babies 12, 22 and 123, who were infected at birth showed a higher level of genetic similarity than over 99.6% of other subset comparisons and much greater than would be expected by chance (black line) (Figure 5—figure supplement 1)” – so is this the explanation why they think there is a genetic bias towards the dominant virus that can get through the transit bottleneck – what might contribute to this bottleneck? Needs further explanation.

FST is a statistical way of identifying genetic similarity between a set of sequences relative to sequences not in that set. The FST test first identified that the genotypes present in the cervices of mothers who transmitted congenitally, and their congenitally infected infants were more similar than could be explained by chance (Figure 5—figure supplement 1). The genes responsible for this high level of genetic similarity were identified in Figure 5. These findings together with demonstration in Figure 4, that only a single genotype is transmitted from mother to baby at any one time, provides evidence that there is generally a bottleneck to maternal-fetal transmission of CMV, whether from breast milk or congenitally, and that certain genotypes may be preferentially transmitted in congenital infection

The y-axis of Figure 5 shows the FST values calculated from this analysis. The higher the FST value, the more significant it is. Each bar represents a gene. We have now increased the font size on the x-axis to make it more legible. The colour of the bars indicates the corresponding p values, with red being the most significant genes.

We know from studies of primary cCMV that maternal viral load is a key factor for transmission. The three mothers transmitting their virus congenitally had higher cervical viral loads than mothers whose babies become infected post-partum (Figure 1). Analysis of data from the whole cohort of mothers confirmed that women who transmitted CMV in utero had mean cervical CMV vial loads at 38 weeks that were 0.83 log_10_ copies/ml (SD=1.0, p=0.02) higher than women who did not transmit CMV in utero (data not shown). We therefore speculate that virus sampled in the cervix is representative of CMV populations that infect and cross the placenta, and that a possible explanation for our findings is that the properties that promote replication to higher titers in genital tissue may also predispose to transplacental infection. Such properties may include increased tropism for and replication in the urogenital tissues and it is possible that the genes identified by FST are contributary.

Alternatively, a number of genes identified in the FST analysis are known to control cell entry and NK cell recognition (the placenta is rich in NK cells) (Table 2). Thus the genotypes that were transmitted congenitally might have beneficial characteristics that allow them to better engage with the placental tissues and hence transmit to the infants.

We have added this to the Discussion.